# Transcriptomics informed discovery of developmentally essential transcription factors

Gillian Forbes[1,2] and Pauline Schaap[1,*]

## ABSTRACT

Transcription factors (TFs) regulate cell differentiation in multicellular organisms and were mostly identified by forward genetics in model organisms. However, genomes contain several-fold more TFs without known roles. To classify these orphans, we investigated conservation, cell-type specificity and temporal expression of the ~290 TFs of Dictyostelia amoebas, which aggregate when starved to form migrating slugs and fruiting bodies consisting of spores and three somatic cell types. Here we deleted seven somatically expressed TF genes and found that four knock-outs were developmentally defective. *ariA⁻* lost slug migration and robust fruiting body formation. *gtaJ⁻* skipped slug migration and directly developed aggregates into robust fruiting bodies. *mybAA⁻* formed multi-tipped aggregates, defective slugs and fruiting bodies with few spores. Hierarchical clustering of the expression profiles of *mybAA* and 45 other multi-tip suppressing genes grouped *mybAA* with seven autophagy genes, with similar developmental defects as *mybAA⁻*, suggesting that *mybAA* induces autophagy gene expression. *mybM⁻* slugs poorly migrated and fruiting bodies had kinked, rough stalks, but normally expressed cell-type marker genes, indicating defective morphogenesis. Overall, transcriptomics informed TF selection proved useful for gene function discovery.

KEY WORDS: Evolution of soma, *ariA*, *gtaJ*, *mybAA*, *mybM*, Life cycle choice, Autophagy, Morphogenetic movement, Social amoebas

## INTRODUCTION

In developing multicellular organisms, the differentiation of cells and their positioning in tissues and organs requires extensive long- and short-range cell signalling. Signals that cause cell differentiation generally activate intracellular pathways that lead to expression of novel gene sets, with the sequence-specific transcription factors (TFs) that control gene expression as the final target. Many TFs with roles in cell differentiation and development were identified through genetic and other approaches. In humans, phenotypes have been associated with mutations in 304 TFs, only 1/5th of the ~1600 TFs in their genomes (Lambert et al., 2018). Understanding the function of the remaining 1300 TFs remains a daunting task.

[1]Division of Molecular Cell and Developmental Biology, School of Life Sciences, University of Dundee, Dundee DD15EH, UK. [2]Institut de Génomique Fonctionnelle de Lyon (IGFL), École Normale Supérieure de Lyon, CNRS, Lyon 69007, France.

*Author for correspondence (p.schaap@dundee.ac.uk)

P.S., 0000-0003-4500-2555

Assigning roles to members of their TF repertoire is more easily achieved for genetically tractable organisms with relatively simple developmental programs. *Dictyostelium discoideum* (*Ddis*) displays facultative multicellularity, where unicellular amoeba feed on soil bacteria, but come together to form freely moving slugs when experiencing starvation. Amoebas in the slug start differentiating into viable spores and three somatic (non-propagating) cell types, the stalk, cup and basal disc cells, which physically lift and support the spore mass (Jermyn et al., 1996; Kin et al., 2022). In addition, the somatic cells use autophagy to provide the spores with metabolites for spore wall synthesis and food storage (Du and Schaap, 2022).

The correct spatio-temporal differentiation of these cell types is determined by environmental stimuli and signals exchanged between the cells, which evoke signal transduction cascades that ultimately regulate TF activity. Currently characterised TFs involved in *Ddis* cell differentiation include: CrtF, SrfA, StkA and SpaA that are required for the formation of normal spores (Chang et al., 1996; Escalante and Sastre, 1998; Mu et al., 2001; Yamada et al., 2018), with SpaA playing the most decisive role; MybC, StatA and Cdl1B that participate in stalk morphogenesis (Guo et al., 1999; Kin et al., 2022; Mohanty et al., 1999); and CudA and BzpF with roles in both spore and stalk differentiation (Fukuzawa et al., 1997; Huang et al., 2011). The TFs MybE, DimB and GtaC contribute to basal disc formation (Keller and Thompson, 2008; Tsujioka et al., 2007; Yamada et al., 2011), while Cdl1A is essential for cup cell differentiation (Kin et al., 2022). The remaining ~22 investigated TFs function earlier in growth or aggregation or regulate cell physiology and cell type proportioning.

Fully assembled genomes (Eichinger et al., 2005; Gloeckner et al., 2016; Heidel et al., 2011) as well as developmental stage- and cell type-specific transcriptomes are available for species representative of the four dictyostelid taxon groups (Eichinger et al., 2005; Gloeckner et al., 2016; Heidel et al., 2011; Kin et al., 2018; Parikh et al., 2010). We previously used this data to retrieve members of all known TF families from species genomes and record the evolutionary conservation of their developmental regulation and cell-type specificity (Forbes et al., 2019). This study detected 440 sequence-specific TFs across 33 families (290 if the less defined AT-hook and C2H2 families are discounted). Initial exploration highlighted Cdl1A, a cup-specific TF that arose through gene duplication in taxon group 4 from Cdl1, the ancestral gene involved in stalk morphogenesis. Cup cells uniquely evolved in group 4, which includes *Ddis*, to support the large spore heads common to group 4. In knock-outs of *cdl1A*, spore heads were no longer lifted and the stalk and cup-specific genes were not expressed, providing the first instance of a genetic change required for evolution of a novel cell-type (Kin et al., 2022).

This result prompted us to use the bioinformatic TF analysis for further TF gene function discovery. Working from observations that *Ddis* TFs, like SpaA, Cdl1A and Cdl1B, that are essential for cell

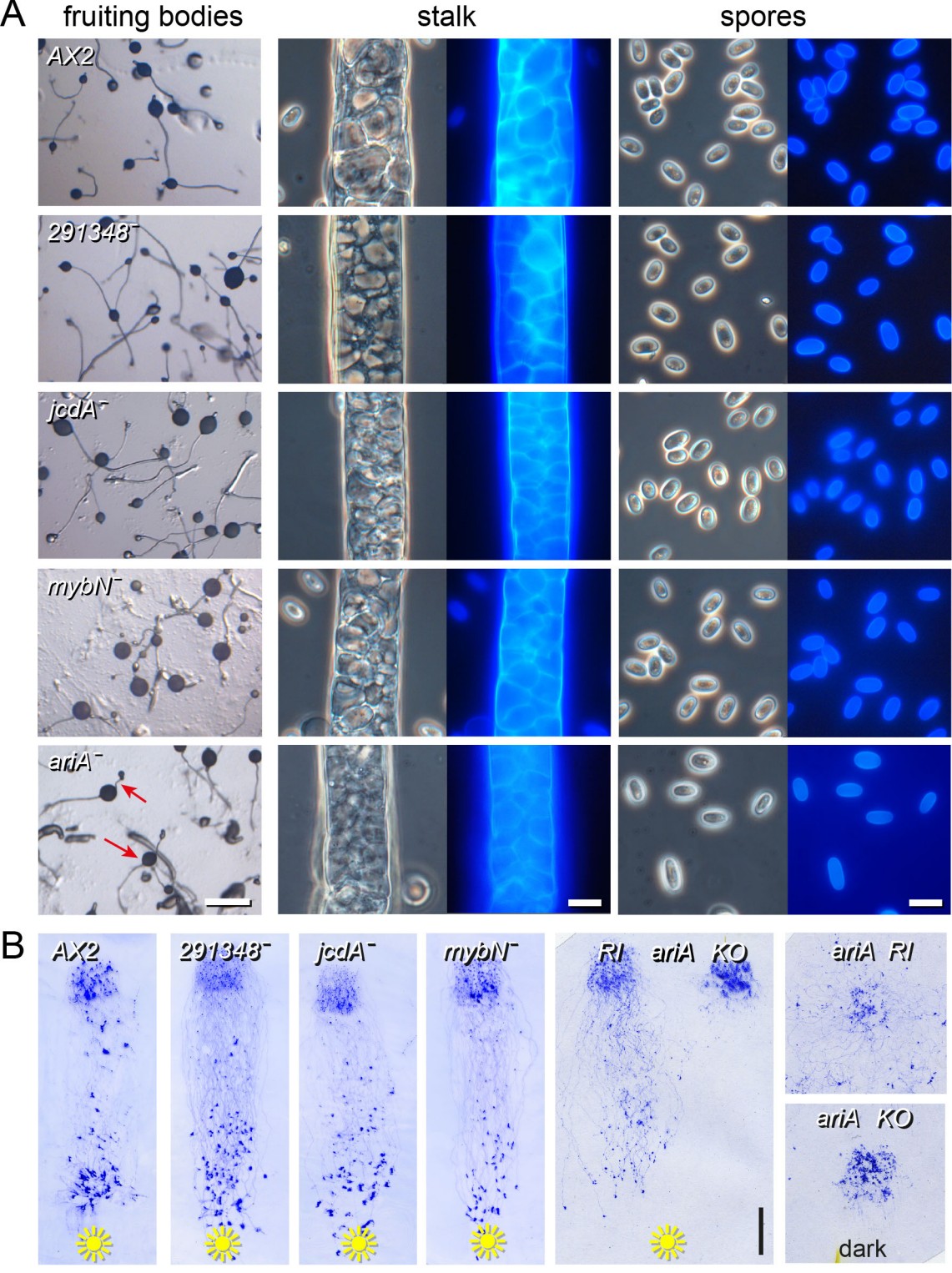

**Fig. 1. Phenotypes of *DDB_G0291348⁻*, *jcdA⁻*, *mybN⁻* and *ariA⁻* mutants.** (A) Development. Lesions in *DDB_G0291348*, *jcdA* or *mybN* were generated in parent strain AX2 by homologous recombination (Fig. S2). Fruiting bodies were imaged directly from growth plates. Red arrows: incompletely lifted spore heads in *ariA⁻* (DDB_G0275333) mutants. Scale bar: 0.5 mm. Stalk and spores from the fruiting structures were lifted into a droplet of 20 µg/ml Calcofluor and imaged under phase-contrast (right) and UV (left). Scale bars: 10 µm. (B) Phototaxis. 3×10⁶ cells grown in axenic medium were spread over a 1 cm² square on 1.5% water agar. Plates were incubated under unilateral light for 3 days. The resulting slugs and slime trails were lifted from the agar on a plastic sheet and stained with Coomassie Blue for 2 min. An *ariA* KO was compared with an RI of the KO vector under both unilateral light and in the dark. Scale bar: 1 cm.

differentiation in a cell-autonomous manner, are upregulated in the cell types that they induce, we deleted seven TF genes enriched in somatic cell types by homologous recombination and found that four of TF knock-outs were developmentally defective. *AriA* is required for slug motility; GtaJ regulates the decision between slug migration and fruiting body formation, MybAA shares phenotypic defects and developmental expression with autophagy genes and may control expression of these genes, while MybM is required for proper stalk morphogenesis. We conclude that a bioinformatics informed approach can assist TF function discovery.

## RESULTS

### TF selection and analysis of mutants from a genome-wide mutagenesis study

To identify TFs that regulate differentiation of the somatic (non-spore) cells of *Dictyostelium*, we screened a comprehensive analysis of TF families across Dictyostelia (Forbes et al., 2019) for genes that showed upregulated and conserved expression in somatic cells. The expression profiles of the seven selected genes are shown in Fig. S1. We generated lesions in the selected genes by homologous recombination with constructs that replace a large fragment of the coding region inclusive of the signature DNA binding domain with the loxP-blasticidin cassette (Faix et al., 2004) (see Fig. S2 for gene knock-out schematics and diagnoses).

The developmental progression of the knock-outs (KOs) was followed with emphasis on post-aggregative development when the genes are normally expressed. No phenotypic abnormalities were discovered in KOs in *DDB_G0291348*, a putative TF with a yeast Gal4 type DNA binding domain (Traven et al., 2006). Mutants in the jumonji and myb type TFs *jcdA* and *mybN* also showed development into well-shaped fruiting bodies, with normal stalk and spore cells with cellulosic walls as evident by Calcofluor staining (Fig. 1A). Directional migration of slugs toward unidirectional light (phototaxis) was also similar to that of the parent strain AX2 (Fig. 1B).

*DDB_G0275333* is an ARID/BRIGHT family TF, further called *ariA*. The *ariA* KO showed a strong slug migration defect, both toward light and in the dark, while a random vector integrant (RI) showed extensive migration under both conditions (Fig. 1B). *ariA⁻* fruiting body formation was asynchronous with some slugs remaining prostrate relatively long, while some structures failed to completely lift the spore-head (Fig. 1A, red arrows). No abnormalities were detected in the *ariA⁻* spores and stalks.

### GtaJ likely controls the slug/fruit switch

The *gtaJ⁻* mutant showed robust aggregation and development into fruiting bodies (Fig. 2A) but its sorogens showed no tendency to migrate either towards light or randomly in the dark (Fig. 2B). Instead, the cells formed fruiting bodies directly at the aggregation centre and almost no prostrate slugs were observed. The few slugs that did fall over did migrate (Fig. 2B, right panel). The *gtaJ⁻* sorogens therefore did not seem incapable of migration but rather skipped it altogether.

Ammonia, which the starving cells produce by proteolysis was shown to prolong slug migration and inhibit initiation of fruiting body formation (Newell et al., 1969; Schindler and Sussman, 1977). To test whether the *gtaJ⁻* cells were less sensitive to ammonia, we transferred tipped mounds to agar with increasing concentrations of NH$_4$Cl. In both parent AX2 and *gtaJ⁻* cells fruiting body formation started to be inhibited at 30 mM NH$_4$Cl and was completely blocked at 100 mM (Fig. 2C). The reluctant migration of *gtaJ⁻* slugs is therefore not due to insensitivity to ammonia. The *gtaJ⁻* phenotype

suggest that in wild-type GtaJ promotes slug migration over culmination, but whether it does so by repressing culmination genes or activating migration genes is at present not clear.

### MybAA regulates aggregate and slug size, sporulation and culmination

The *mybAA* knockouts initially aggregated normally with inflowing streams, but in late aggregation, streams detached from the centre to form smaller mounds. The central mound also split into many small slugs by forming multiple organizing tips (Fig. 3A, Movie 1). The slugs migrated about 5 h longer than wild-type slugs, while making aborted attempts to culminate. Newly formed fruiting bodies often toppled over, with the smaller structures being more successful to remain erect. Fruiting body stalks were thin but showed normal cellulose deposition when stained with Calcofluor (Fig. 3B). Spore walls stained normally, but there was a larger proportion of round spores (Fig. 3B) and the efficiency of spore production was five-fold reduced (Fig. 3C). The *mybAA⁻* stalks also lacked an obvious basal disc (Fig. 3D), which, with the thin stalks, probably accounted for fruiting body prostration. The *mybAA⁻* slugs oriented towards light but migrated poorly (Fig. 3E). Migrating slugs left vacuolated cells with cellulose walls behind in the slime trail as well as larger cell clumps with both undifferentiated and vacuolated cells (Fig. 3F). The multi-tipped phenotype of *mybAA⁻* aggregates was not rescued by chimeric development with 10% wild-type cells (Fig. 3G) indicating that this defect is cell-autonomous.

Multiple tip formation is not a rare mutant phenotype and to investigate relationships between *mybAA⁻* and other multi-tipped mutants, we performed hierarchical cluster analysis on the developmental- and cell type-specific expression profiles of the mutant genes. Fig. S3 shows that *mybAA* clusters with seven autophagy genes and four unrelated genes, indicative that *mybAA* possibly regulates expression of autophagy genes.

### MybM controls stalk morphogenesis and slug migration

*mybM⁻* cells developed normally until culmination when the spore mass was elevated from the substratum by the newly formed stalk. However, the *mybM⁻* stalks were thicker and much shorter than those of the parent and showed a distinct kink. The kink mostly occurred shortly below the tip, giving the spore mass a triangular shape (Fig. 4A). Calcofluor staining showed proper cellulose deposition in the spore and stalk walls, but the surface of the *mybM⁻* stalk tube was less smooth with the stalk cells bulging out (Fig. 4B). The upper stalk was actually kinked multiple times inside the spore head (Fig. 4C) and the spore mass had likely become trapped in the kinks.

In addition to the stalk defects, *mybM⁻* also showed reduced slug migration in the phototaxis assay, although the few migrating slugs still oriented towards the light (Fig. 4D). Chimeric development with 10% wild-type cells did not restore normal fruiting body formation (Fig. 4E), indicating that the stalk defect is cell-autonomous.

To investigate cell differentiation, *mybM⁻* and parent AX2 cells were transformed with the prestalk and stalk marker *ecmA-lacZ*, the stalk marker *staH (DDB_G0278745)-lacZ* (Forbes et al., 2022), the cup marker *beiG-lacZ* and the prespore markers *cotC-lacZ* and *psA-lacZ*. Migrating slugs, obtained by overnight development on water agar under unilateral light, and fruiting bodies, obtained by development on non-nutrient (NN) agar, were stained with X-gal. Both *mybM⁻* and AX2 slugs expressed *ecmA* at the anterior of the slug and *psA* and *cotC* at the posterior, but the expression domain of *psA* appeared smaller in *mybM⁻* slugs (Fig. S4A). The relative areas that expressed *psA*, *cotC* and *ecmA* across the slug were therefore

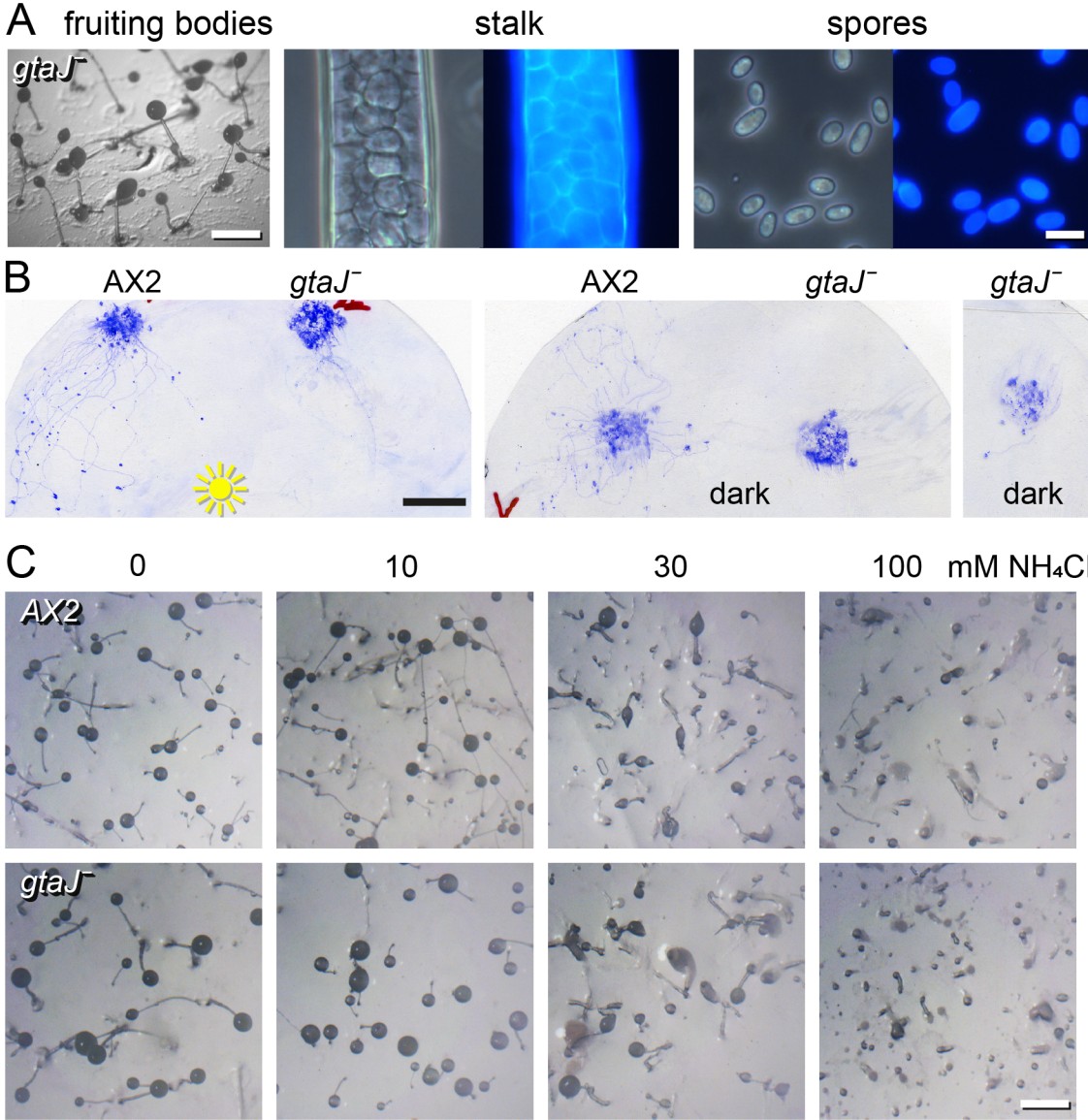

**Fig. 2. *gtaJ* knockouts lack slug migration.** (A) Development. *gtaJ⁻* fruiting bodies, stalks and spores were imaged as in Fig. 1A. Left scale bar: 0.5 mm, right scale bar: 10 µm. (B) Slug migration. Parent AX2 and *gtaJ⁻* cells were plated as 10 µl droplets of 3×10⁸ cells on water agar and incubated either under unilateral light or in the dark for 2 days. Slugs and slime trails were stained with Coomassie Blue as in Fig. 1B. Right panel: rare instance of *gtaJ⁻* slug migration. Scale bar: 1 cm. (C) Ammonia sensitivity. AX2 and *gtaJ⁻* cells were developed on dialysis membrane supported by NN agar until tipped mounds had formed and membranes were then transferred to NN agar of pH 7.4 with the indicated concentrations of NH₄Cl. Structures were imaged after 24 h at 21°C. Scale bar: 0.5 mm. A second experiment showed a similar result.

measured and showed a trend of the *ecmA* expressing prestalk region to be somewhat enlarged and the *psA* and *cotC* expressing prespore regions to be reduced. However, only for *psA* did the reduced expression in *mybM⁻* reach statistical significance (Fig. S4B).

In fruiting bodies, both *ecmA* and *staH* were well expressed in the AX2 and *mybM⁻* stalks with the kink in the latter's stalk clearly apparent (Fig. S4C). Expression of *beiG* was re-oriented, with the lower cup no longer at the base but at the side of the spore head, due to the cup cells aligning to the kinked stalk. *ecmA*, which in AX2 is both expressed in the stalk and upper and lower cups, also showed disoriented expression in the *mybM⁻* spore head. The relatively normal stalk expression patterns in *mybM⁻* suggests that their kinked stalks are not due to defective stalk gene expression.

Finally, we compared the sporulation efficiency of *mybM⁻* to that of AX2 and found that *mybM⁻* produced about 30% less spores from

a known amount of plated cells (Fig. S4D), which may be related to the reduced area of prespore gene expression in *mybM⁻* slugs.

## DISCUSSION

Dictyostelia contain 440 sequence-specific TFs (290, if less well defined AT-hook and C2H2 families are discounted), but a biological role for only 34 was known in 2019 (Forbes et al., 2019) and for four more later (Hao et al., 2024; Kin et al., 2022). Here, we used stage- and cell-type specific transcriptome data to select seven deeply conserved TFs with putative roles in the differentiation of somatic cells, which physically and metabolically support the propagating spores (Du and Schaap, 2022). Large segments of their genes, including their signature DNA binding domains were deleted by homologous recombination and their phenotypes were examined.

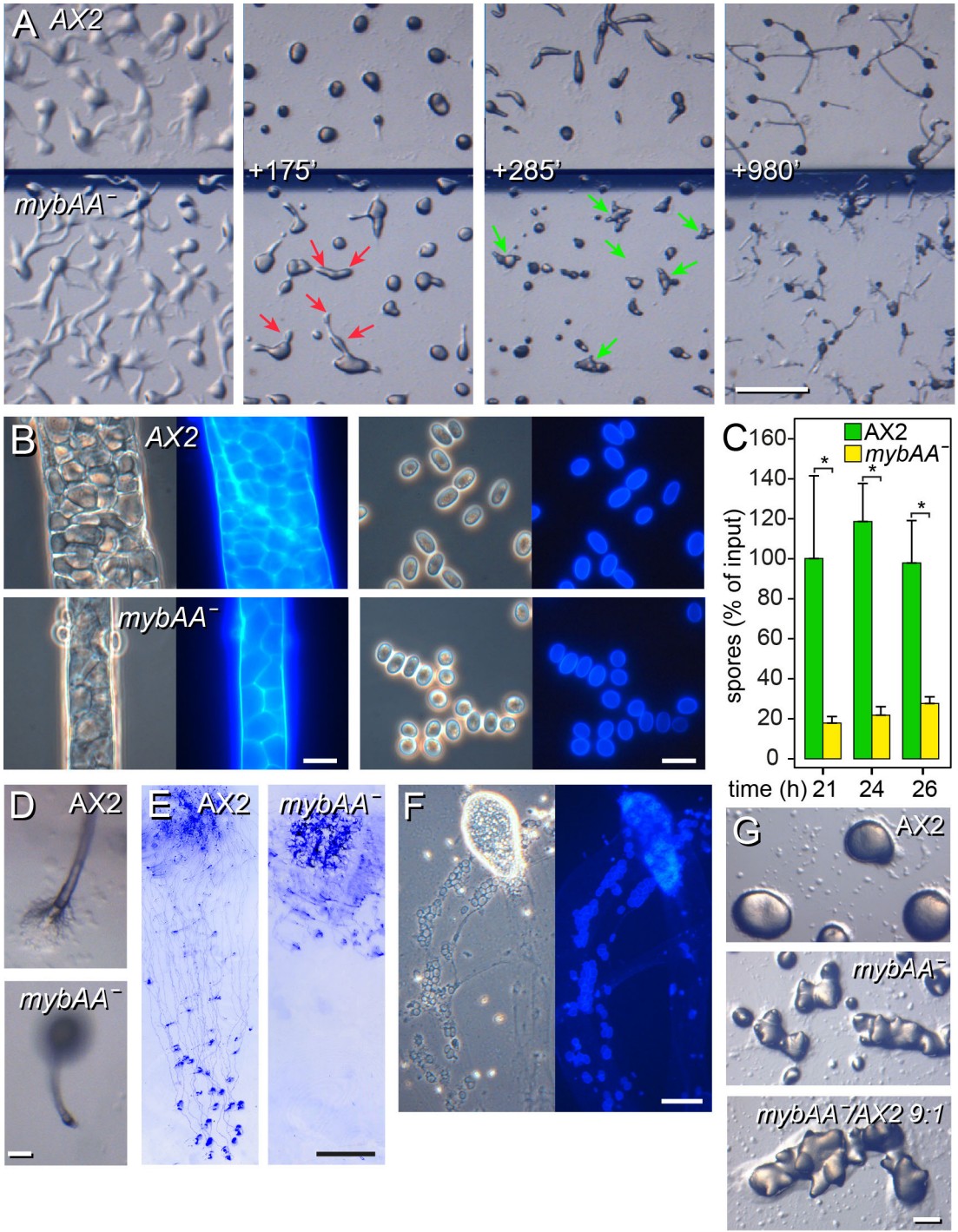

**Fig. 3. Fragmentation of *mybAA⁻* aggregates and slugs.** (A) Development. AX2 and *mybAA⁻* cells were developed on NN agar at $1.5×10^6$ cells/cm², separated by a plastic barrier, and imaged by time-lapse microscopy (Movie 1). Images taken at aggregation and the indicated minutes later are shown. Red and green arrows indicate streams splitting up and multiple tip formation, respectively. Scale bar: 1 mm. (B) Stalk and spores. Fruiting bodies were transferred to slides, stained with Calcofluor and imaged as described for Fig. 1. Scale bars: 10 µm. (C) Sporulation efficiency. $3×10^6$ cells were plated on 1 cm² filters and developed into fruiting bodies for the indicated periods, when spores were washed off and counted. Data are expressed as percentage of counted cells and represent means and standard deviations from three experiments with four replicate counts. Significant differences between AX2 and *mybAA⁻* are indicated by * for $P<0.001$. (D) Basal disc. The stalk base of AX2 and *mybAA⁻* fruiting bodies, developed on NN agar. Scale bars: 50 µm. (E) Phototaxis. AX2 and *mybAA⁻* cells, harvested from axenic media, were plated on water agar, incubated under unilateral light for 3 days and stained with Coomassie Blue as described in the Materials and Methods. Scale bar: 1 cm. (F) Slime trail. *mybAA⁻* migrating slugs and slime trails were stained with Calcofluor and imaged *in situ* on NN agar. Scale bar: 50 µm. (G) Chimeric development. *mybAA⁻* and AX2 cells were developed separately and as a 9:1 mixture at $3×10^6$ cells/cm² on NN agar and imaged at the mound stage. Scale bar: 0.2 mm.

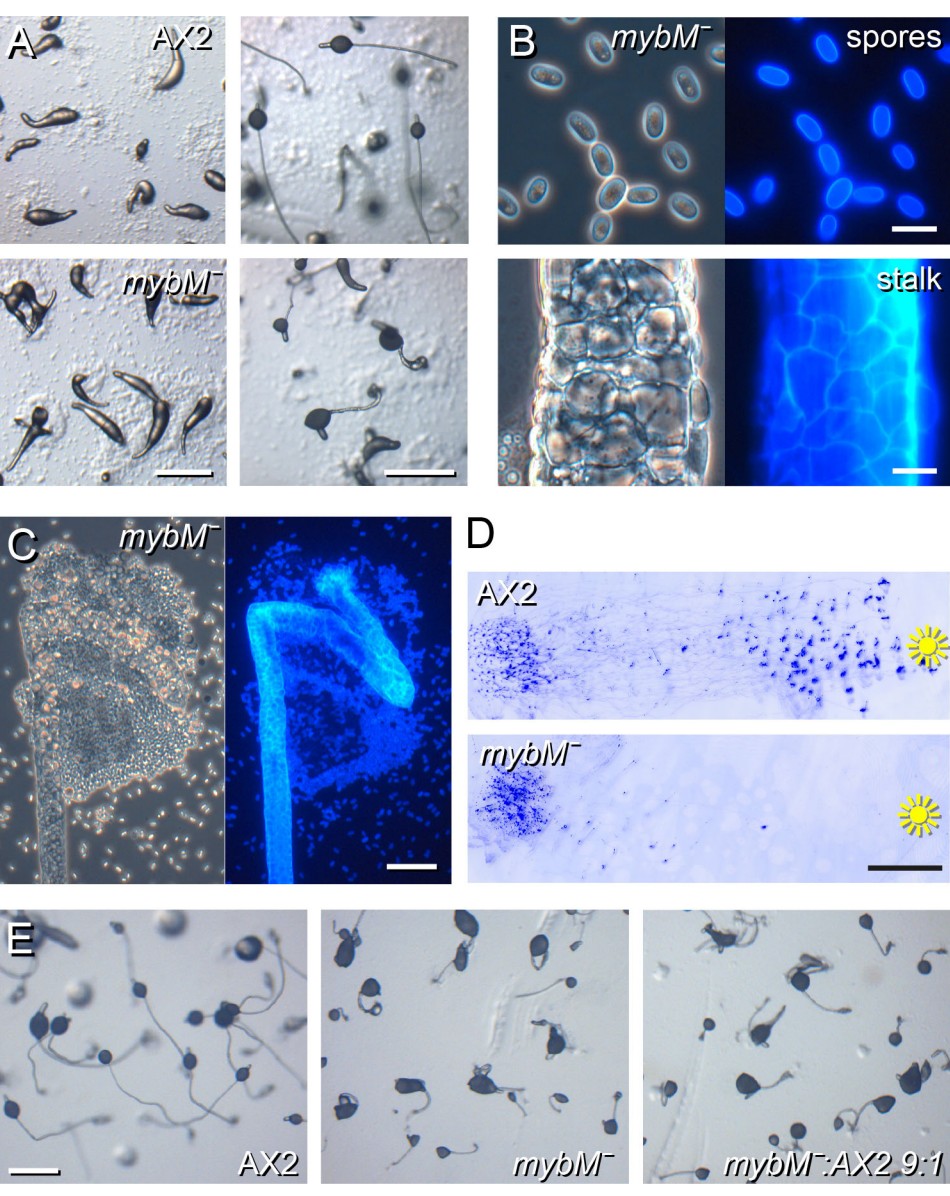

**Fig. 4. *MybM⁻* has stalk and slug migration defects.** (A) Development. AX2 and *mybM⁻* cells were developed on NN agar at 3×10⁶ cells/cm² and imaged at the slug and fruiting body stage. Scale bar: 0.5 mm. (B,C) Stalk, spores and spore head. Fruiting bodies were transferred to slides and stained with Calcofluor. (B) Spores and stalk, scale bar: 10 µm. (C) Spore head, scale bar: 50 µm. (D) Phototaxis. AX2 and *mybM⁻* cells were incubated under unilateral light for 3 days and stained with Coomassie Blue as described in the Materials and Methods. Scale bar: 1 cm. (E) Chimeric development. AX2 and *mybM⁻* cells were developed separately and as a 1:9 mixture at 3×10⁶ cells/cm² on NN agar and imaged at the fruiting body stage. Scale bar: 0.5 mm.

KOs in three (*DDB_G0291348*, *jcdA* and *mybN*) of the seven selected genes showed no discernible aberration in development to fruiting bodies, spore or stalk morphology or in slug migration and phototaxis. This does not necessarily mean that these TFs have no function, but that their function either overlaps with that of other TFs, affects processes that require detailed investigation or are not affected under laboratory conditions.

### Four mutants showed slug motility defects

The KO of ARID/BRIGHT TF *ariA*, showed strongly reduced motility of prostrate slugs and mildly defective and asynchronous fruiting body formation, with some slugs delaying culmination and some spore heads incompletely uplifted. This points to an underlying problem in the coordinated cell movements that enable slug motility and fruiting body morphogenesis.

KOs in *gtaJ* also lacked slug migration under unilateral light or in the dark (Fig. 2) but otherwise displayed very robust aggregation and development into large well-shaped fruiting bodies. Here, the fruiting bodies formed directly from aggregates and early sorogens hardly ever fell over to enter slug migration. This suggest that GtaJ

determines the choice of sorogens to enter migration rather than fruiting body erection.

Several factors such as absence of incident light, low ionic strength of the substratum, high humidity, and presence of secreted metabolites promote the decision to migrate rather than erect fruiting bodies (Newell et al., 1969). The first three of those are routinely employed by testing slug migration on fresh water agar in unilateral light.

The secreted metabolite is ammonia (Schindler and Sussman, 1977), which activates the histidine kinase DhkC (Singleton et al., 1998). DhkC in turn phosphorylates and thereby activates the cAMP phosphodiesterase RegA, decreasing intracellular cAMP levels and inhibiting PKA. PKA activity is essential for stalk cell differentiation and thereby fruiting body formation (Harwood et al., 1992). The *gtaJ⁻* phenotype resembles that of cells lacking *amtA*, a putative ammonia sensor (Singleton et al., 2006). However, *amtA⁻* sorogens no longer show inhibition of fruiting body formation by high ammonia, whereas *gtaJ⁻* sorogens showed similar inhibition by ammonia as wild-type cells (Fig. 2C). This suggests that GtaJ does not regulate expression of components in the ammonia

pathway and acts on other genes that promote slug migration over culmination.

Negative effects of lack of GtaJ on sporulation were not evident during development on agar. However, in their natural habitat, phototaxis brings migrating slugs to the top layer of their soil substratum, where spores are more readily dispersed. Here lack of GtaJ and thereby slug migration would negatively affect *Ddis* propagation.

For *mybAA⁻* and *mybM⁻* mutants defective slug motility is part of a larger complex of morphogenetic defects and, as with *ariA⁻*, the actual cause of the defect remains unclear.

### MybAA may positively regulate expression of autophagy genes

Deletion of *mybAA* caused a spectrum of defects. While initially forming large streaming aggregates like its parent AX2, aggregation streams split up into smaller mounds and mounds formed multiple tips resulting in many small slugs. The slugs showed reduced migration and left cell clumps and vacuolated cells behind, while many failed to erect fruiting bodies. The residual small fruiting bodies had thin stalks that lacked a basal disk and often collapsed. Spores were smaller and less elliptical than those of AX2 and there was an 80% reduction in spore production. Since slug migration is positively correlated with slug length (Bonner, 1995), part of the migration defect could be a consequence of the aggregate fragmentation. However, the cause of the reduced sporulation efficiency and aberrant spore morphology is less easily explained.

Aggregate fragmentation also occurred in a *smlA⁻* mutant (Brock and Gomer, 1999), which oversecretes counting factor (CF), a complex of proteins that limit aggregate size. In mutants lacking each of the CF components, aggregate streams do not break up, resulting in very large fruiting structures (Brock et al., 2003). CF likely acts by decreasing cell-cell adhesion and increasing cell motility in aggregation streams, thus impacting their coherence (Roisin-Bouffay et al., 2000; Tang et al., 2002).

While MybAA could potentially act to activate *smlA* transcription, the *smlA⁻* phenotype does not incorporate the formation of multiple tips on aggregates, which in *mybAA⁻* is the more prominent cause of reduction in slug size. A multi-tipped phenotype has been observed in mutants in 45 genes, many of which have no obvious relationship to each other. Hierarchical clustering of the developmental and cell-type specific expression profiles of these genes with *mybAA*, highlighted a 12 gene cluster that contained *mybAA* and seven genes (*atg5*, *atg6*, *atg8*, *atg9*, *atg12*, *tipD/atg16* and *atg101*) with roles in autophagy.

Autophagy mutants also show complex phenotypes. For instance, *atg6⁻*, *atg8⁻*, *atg12⁻*, *atg101⁻*, *tipD/atg16⁻* mutants form multi-tipped aggregates and small fruiting bodies with few spores (Mesquita et al., 2017; Otto et al., 2004). Slugs migrated but left parts behind in *atg12⁻*, and *tipD/atg16⁻* mutants, while spore viability was strongly reduced (Fischer et al., 2019). *atg5⁻*, *atg7⁻* and *atg9⁻* mutants also lacked slug migration, overproduced stalk cells, produced few spores and lacked cAMP induction of prespore gene expression (Yamada and Schaap, 2019).

The *mybAA⁻* phenotype combines the defects found in autophagy mutants but is less severe. The co-expression of *mybAA* with preferentially autophagy genes suggests that it acts as a TF enhancing autophagy gene expression, but likely in an overlapping role with another TF.

### MybM is required for stalk morphogenesis

Cells lacking *mybM* formed fruiting bodies with short, kinked stalks (Fig. 4). This phenotype was reminiscent of KOs in the interacting autophagy genes *knkA* and *bcas3* (Yamada and Schaap, 2021). However, unlike *knkA⁻* and *bcas3⁻*, which form almost no walled spores, *mybM⁻* formed normal viable spores, although 20% less than AX2. Slugs were phototactic but did not migrate very far. The outline of the relatively thick stalks was rough with cells bulging out and the upper stalk was often kinked multiple times, giving the impression that the upper stalk attempted to extend as normal but folded under the weight of the spore head. There was a slight increase in the prestalk over prespore regions of slugs, but stalk and cup markers genes were normally expressed. The kinked phenotype is therefore unlikely to be caused by defective cell differentiation.

The stalk owes its rigidity to the cellulose stalk tube and turgor pressure of the stalk cell contents against their cellulose walls. The developing stalk is further constrained by adherence junctions between the surrounding prestalk cells that are connected by actin bundles, thus forming an actin ring around the stalk tube (Grimson et al., 2000). Lack of the junction component β-catenin or regulators of the actin cytoskeleton could therefore also cause stalk defects. The poor migration of *mybM⁻* slugs slightly favours defective actin polymerization over defective cellulose deposition as a probable cause.

## MATERIALS AND METHODS

### Cell culture and development
Cells were grown on full or 1/5th SM agar (Formedium, UK) in association with *Klebsiella aerogenes* or in HL5 (Formedium) in culture dishes or shaking culture (150 rpm) at 22°C. For development, cells were harvested at the exponential phase, washed with KK2 (16 mM $KH_2PO_4$, 4 mM $K_2HPO_4$, pH 6.5) and plated at $3×10^6$ cells/cm² on NN agar (1.5% agar in 8.8 mM $KH_2PO_4$ and 2.7 mM $Na_2HPO_4$).

### DNA constructs and transformation
Sequences flanking essential domains of the gene of interest (Fig. S2) were amplified by PCR using oligonucleotide primers listed in Table S1. Amplified sequences were subcloned into pCR4-TOPO_TA (Thermo Fisher Scientific) and validated by DNA sequencing. The fragments were excised and ligated into vector pLPBLP (Faix et al., 2004), using restriction sites that were included in the oligonucleotide primers. The KO construct was excised and introduced into *Ddis* Ax2 by electroporation. Transformed cells were selected by including 10 µg/ml blasticidin in the HL5 growth medium and cloned out on SM agar plates with *K. aerogenes*. Genomic DNAs were isolated from individual clones and tested for gene KO by two or three sets of PCR reactions (Fig. S2), using primers listed in Table S1.

### Imaging and time lapse movies of development
Developmental structures, developed on NN agar or on 1/5th SM growth plates, were imaged under transillumination with a Leica MZ16 stereo microscope, MicroPublisher 3.3 RTV camera and Qcapture Pro software. To image spore and stalk cells, fruiting bodies were lifted into a droplet of 20 µg/ml Calcofluor White (Sigma-Aldrich) on a glass slide and overlaid with a coverslip. Spores and stalks were imaged under phase-contrast and ultraviolet light.

For time lapse movies, mutant and parent cells were plated at $1.5×10^6$ cells/cm² on NN agar, separated by a narrow strip of overhead sheet inserted in the agar. Images were taken every 5 min until fruiting structures had formed, using the Leica stereomicroscope, and combined with Qcapture Pro at 10 frames per second.

### Slug migration
Exponentially growing cells were harvested from HL5 shaking culture, washed twice with water, resuspended to $10^8$ cells/ml and plated on water agar (1.5% agar in water) either as a 10 µl droplet or as 30 µl spread over 1 cm². Plates were incubated under unidirectional light or in the dark for 2 or 3 days. Next, slugs and slug trails were lifted onto a plastic overhead sheet

and stained for 2 min with Coomassie Brilliant Blue (0.3% Coomassie Brilliant Blue R-250, 50% methanol, 10% acetic acid). The sheet was washed with water and dried and then imaged using a document scanner.

## Sporulation efficiency

Fruiting bodies were developed from $3\times10^6$ cells, spread on a $1\text{ cm}^2$ nitrocellulose filter, supported by NN agar. Filters were vortexed in 1 ml of 0.1% Triton-X100 and spores were counted four times for each filter using a hemacytometer. Spore numbers were expressed as percentage of the number of cells plated.

## Cell-type marker gene expression

Parent and KO cells were transformed with fusion constructs of the *lacZ* reporter gene with cell-type specific promoters and selected at 50 µg/ml G418, unless described otherwise. For fruiting bodies, transformed cells were harvested at the exponential stage and plated on nitrocellulose membranes supported by NN agar at $3\times10^6$ cells/cm² and incubated at 22°C for 24 h. For migrating slugs, membranes were incubated overnight on water agar under unilateral light. Structures were fixed in 0.5% glutaraldehyde and stained with X-gal (Kin et al., 2022) with parent and KO stained equally long. To assess relative prestalk and prespore regions, stained and unstained areas of slugs were measured using the measure feature of ImageJ (Collins, 2007) and expressed as fraction of the full slug area.

## Acknowledgements
We thank MRC PPU DNA Sequencing and Services for DNA sequence validation and the School of Life Sciences Central Technical Services for preparation of culture media.

## Competing interests
The authors declare no competing or financial interests.

## Author contributions
Conceptualization: G.F.; Data curation: G.F.; Formal analysis: G.F., P.S.; Funding acquisition: P.S.; Investigation: G.F.; Supervision: P.S.; Validation: P.S.; Writing – original draft: G.F.; Writing – review & editing: P.S.

## Funding
This research was funded by grant 742288 of the European Research Council. Open Access funding was provided by European Research Council. Deposited in PMC for immediate release, distributed under Open Access CC-BY license.

## Data and resource availability
All DNA constructs and knock-out mutants generated in this work were deposited in the *Dictyostelium* Stock Center http://dictybase.org/StockCenter/StockCenter.html. All relevant data and details of resources can be found within the article and its supplementary information.

## Peer review history
The peer review history is available online at https://journals.biologists.com/bio/lookup/doi/10.1242/bio.062354.reviewer-comments.pdf

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
