## [Peer Review File · Biology Open]

Transcriptomics informed discovery of developmentally essential transcription factors

Gillian Forbes and Pauline Schaap

DOI: 10.1242/bio.062354

Editor: Tristan Rodríguez

Review timeline

Submission to sister journal: 3 September 2025

Editorial decision at sister journal: 30 October 2025

Transfer to Biology Open: 3 November 2025

Accepted: 4 November 2025

Original submission to sister journal

First decision letter

MS Title: TRANSCRIPTOMICS INFORMED DISCOVERY OF DEVELOPMENTALLY ESSENTIAL TRANSCRIPTION FACTORS

Authors: Gillian Forbes; Pauline Schaap

Article Type: Research Article

I have now received all the referees' reports on the above manuscript, and have reached a decision. I am sorry to say that the outcome is not a positive one. The referees' comments are appended below.

As you will see, the referees raise some significant concerns about your paper, and are not strongly in favour of publication. Having looked at the manuscript myself, I agree with their views, and I must therefore, reject your paper.

I do realise this is disappointing news, but we receive many more papers than we can publish, and we can only accept manuscripts that receive strong support from referees.

I do hope you find the comments of the referees helpful, and that this decision will not dissuade you from considering our journal for publication of your future work. Many thanks for sending your manuscript to us.

Reviewer 1

Comments for the author

SUMMARY OF THE ADVANCE MADE IN THIS PAPER AND ITS POTENTIAL SIGNIFICANCE TO THE FIELD

This paper is a proof of principle test of a bioinformatics approach to identifying transcription factors important in *Dictyostelium* development. From previously published bioinformatic analysis, seven TFs were selected as likely to have a role in development and the genes knocked out. The phenotypes of the resulting mutants were characterised at a basic descriptive level, such as: developmental progression, slug formation and migration, phototaxis and terminal differentiation of stalk and spore cells. Two mutants developed normally, while the other five had varied and quite complex developmental phenotypes, not strongly related to each other. Perhaps the most

interesting gene was mybAA, where mutation caused a multi-tipped phenotype like some autophagy mutants and whose expression clusters with that of autophagic genes. It was suggested that it might control autophagic gene expression, although this was not tested. The work did not progress to RNAseq of the mutants, or identifying TF binding sites or testing the response of the mutants to developmental inducers. At a formal level, one could argue that the initial hypothesis is not properly tested because we do not know what proportion of TF mutants selected at random would also yield developmental phenotypes.

The work is of a high technical quality, is useful for increasing our understanding of Dictyostelium development and should be published in a journal of record, but in my opinion, it yields too little insight into development for publication in Development.

Reviewer 2

Comments for the author

SUMMARY OF THE ADVANCE MADE IN THIS PAPER AND ITS POTENTIAL SIGNIFICANCE TO THE FIELD

This paper uses transcriptomics data to identify putative TFs with developmental roles in Dictyostelium. This is a reasonable approach, although at the scale it is carried out at, the TF selection seems a little ad hoc. If there are several hundred TFs, why only pick 7. The choice could be better motivated. I am not sure the approach is in itself especially novel. Although it is difficult to pick a specific example, pulling candidate genes from RNAseq seems to be a fairly standard practice across many different developmental model systems. I think the data are very well presented for the different TF phenotypes, and it is interesting that even though there are several hundred TFs, more than 50% had clear effects on development, despite any expectation of redundancy. The deeper exploration of the specific phenotypes could have ameliorated my concerns, however, these studies seemed to stop before that point. For example, the mybAA phenotype has a very similar phenotype to the lagC mutants (also called tgrB and C) in addition to others, yet this has not been explored. There is discussion of autophagy in relation to mybAA, but again this is not explored. Overall, I feel the work is very well carried out and presented, and it will be useful to have these phenotypes catalogued but I do not see substantial technological or conceptual progress as the data are presented.

SUGGESTIONS TO AUTHORS

I would feel at the current state of progress that an exploration of a complete family (eg. gta), or a more comprehensive investigation of one or two TFs might be more contemporary. For example we know that several TFs shuttle- including mybs and gtas, but there is no exploration of that here. What are the effects on specific target genes- do they bind specific target genes?

Reviewer 3

Comments for the author

SUMMARY OF THE ADVANCE MADE IN THIS PAPER AND ITS POTENTIAL SIGNIFICANCE TO THE FIELD

This study provides a systematic functional analysis of developmentally regulated TFs in Dictyostelium by integrating evolutionary conservation data with targeted gene knockouts. By identifying and characterizing multiple TF mutants, including ariA⁻, gtaJ⁻, mybAA⁻, and mybM⁻, the authors establish new connections between specific TFs and key developmental processes such as cell differentiation, morphogenesis, and the transition from slug migration to fruiting body formation. This work significantly expands the functional landscape of Dictyostelium TFs and offers a valuable foundation for future mechanistic studies of transcriptional regulation in multicellular development.

SUGGESTIONS TO AUTHORS

While this study provides a valuable function for understanding transcriptional control in *Dictyostelium* development, it currently remains a descriptive level and does not yet establish mechanistic links between TF activity and developmental processes. The current interpretation relies mainly on transcript expression levels, but transcription factors are often regulated through post-translational modifications, nuclear translocation, or protein-protein interactions. To meet the journal's emphasis on advancing our understanding of developmental mechanisms, additional experiments that uncover regulatory pathways or downstream targets would be beneficial.

In particular, the processes by which MybM influences stalk morphogenesis and whether MybAA directly regulates autophagy-related genes are not experimentally addressed. Additional analyses, such as promoter-reporter assays, CHIP-based binding studies, or transcriptomic profiling of the mutant strains could help clarify these mechanisms and substantially strengthen the manuscript.

Minor comments:

1. The term "somatic cell types" is not yet widely established in *Dictyostelium* research, and readers may find it somewhat unfamiliar. To improve clarity, it may be helpful to define the term explicitly, for example by adding "(non-spore)" as in the beginning of the Results section, or by briefly describing which cell types are included in this category.
2. In some of the figure legends, the scale bar unit appears to be written as "mM" instead of "mm."
3. In Figure 3, the panel label "G" appears to be missing in the figure itself.
4. The line breaks for the Yamada, Y. and Schaap, P. references seem misformatted.
5. In Figure S1 & S3, it would be helpful to clarify whether the multiple blocks shown under spores and stalk (for *D. discoideum*) represent different developmental time points or independent RNA-seq datasets.
6. In Figure S1, while the authors have long used the former name (*P. pallidum*), briefly noting the current taxonomy could help avoid confusion among readers.

Transfer to Biology Open

Author response to reviewers' comments

Response to reviewers comments

All three reviewers agree that the study significantly expands our knowledge of transcription factors that regulate important aspects of *Dictyostelium* development. However, they also argue that the work does not go far enough into establishing the mechanisms by which the four developmentally essential transcription factors exert their function and the repertoires of genes that are controlled by each individual transcription factor.

This was not the intended aim of the study, which was to use available stage- and cell-type specific transcriptome data across the *Dictyostelium* phylogeny to select conserved TFs with putative roles in the somatic cell differentiation.

The additional experiments proposed by e.g. reviewer 1 - RNAseq of the mutants, identification of TF binding sites and testing responses of the mutants to developmental inducers represent years of additional work. The results would likely require publication in separate manuscripts due to the processes regulated by the transcription factors being quite divergent.

Termination of funding and transition to emeritus status of the principal investigator makes further extensive experimentation impossible.

The reviewers agree that the research is of high technical quality and should be published in a journal of record. There are no specific criticism on the experimentation or conclusions drawn from the data, apart from a number of minor comments by reviewer 3. These comments were all addressed and are shown as highlights in the revised version of the manuscript.

Reviewer 3 Minor comments:

1. The term "somatic cell types" is not yet widely established in Dictyostelium research, and readers may find it somewhat unfamiliar. To improve clarity, it may be helpful to define the term explicitly, for example by adding "(non-spore)" as in the beginning of the Results section, or by briefly describing which cell types are included in this category.

AU: corrected - 2nd paragraph of Introduction

2. In some of the figure legends, the scale bar unit appears to be written as "mM" instead of "mm."

AU: corrected - highlights in legends to figs 1,3 and 4

3. In Figure 3, the panel label "G" appears to be missing in the figure itself.

AU: corrected

4. The line breaks for the Yamada, Y. and Schaap, P. references seem misformatted.

AU: corrected - last entry in reference section

5. In Figure S1 & S3, it would be helpful to clarify whether the multiple blocks shown under spores and stalk (for *D. discoideum*) represent different developmental time points or independent RNA-seq datasets.

AU: corrected - see highlighted text in the legends to Figs S1 and S3

6. In Figure S1, while the authors have long used the former name (*P. pallidum*), briefly noting the current taxonomy could help avoid confusion among readers.

AU: corrected aka *Heterostelium album* is added to *P. pallidum* in legend to fig. S1

First decision letter

MS ID: bio.062354

MS Title: TRANSCRIPTOMICS INFORMED DISCOVERY OF DEVELOPMENTALLY ESSENTIAL TRANSCRIPTION FACTORS

Authors: Gillian Forbes; Pauline Schaap

Article Type: Research Article

I am happy to tell you that your manuscript has been accepted for publication in Biology Open, pending our standard publication integrity checks. It was accepted on 4th November 2025.